# Consequences and Resolution of Transcription–Replication Conflicts

**DOI:** 10.3390/life11070637

**Published:** 2021-06-30

**Authors:** Maxime Lalonde, Manuel Trauner, Marcel Werner, Stephan Hamperl

**Affiliations:** Institute of Epigenetics and Stem Cells (IES), Helmholtz Zentrum München, 81377 Munich, Germany; maxime.lalonde@helmholtz-muenchen.de (M.L.); manuel.trauner@helmholtz-muenchen.de (M.T.); marcel.werner@helmholtz-muenchen.de (M.W.)

**Keywords:** transcription–replication conflicts, genomic instability, R-loops, torsional stress, common fragile sites, early replicating fragile sites, replication stress, chromatin, fork reversal, MIDAS, G-MiDS

## Abstract

Transcription–replication conflicts occur when the two critical cellular machineries responsible for gene expression and genome duplication collide with each other on the same genomic location. Although both prokaryotic and eukaryotic cells have evolved multiple mechanisms to coordinate these processes on individual chromosomes, it is now clear that conflicts can arise due to aberrant transcription regulation and premature proliferation, leading to DNA replication stress and genomic instability. As both are considered hallmarks of aging and human diseases such as cancer, understanding the cellular consequences of conflicts is of paramount importance. In this article, we summarize our current knowledge on where and when collisions occur and how these encounters affect the genome and chromatin landscape of cells. Finally, we conclude with the different cellular pathways and multiple mechanisms that cells have put in place at conflict sites to ensure the resolution of conflicts and accurate genome duplication.

## 1. Introduction

Transcription and replication are the two major nuclear processes that use large cellular resources to allow gene expression and DNA duplication, respectively. In particular, RNA Polymerase II (RNAP II) transcribes protein-coding and non-coding genes as well as other large regions of the genome in a pervasive manner [1]. At the same time, DNA replication forks must access and duplicate every single base pair of the genome in the short time window of the S-phase. Thus, in this critical cell cycle stage, tight regulation and coordination of these two processes are strictly required to avoid interference, which would otherwise lead to encounters of the transcription and replication machineries on the same DNA template.

One conceivable solution to avoid such transcription–replication conflicts (TRCs) could be to restrict transcription outside of the S-phase, thereby excluding potential interference with DNA replication. However, several gene sets display crucial S-phase-specific functions, for example, replication factors and core histone genes that together allow assembly of the newly synthesized DNA into nucleosomes [2,3], ribosomal RNA genes to provide a continuous supply of ribosomes [4] or other long genes that are initiated in the G1 phase, but completion of their transcription cycle extends into the S-phase [5].

Early microscopy studies of active sites of transcription and replication revealed spatial segregation of the two processes in S-phase nuclei [6], albeit to different degrees in human cancer cell lines [7,8]. As both transcription and replication appear within discrete nuclear foci containing high concentrations of RNAP complexes or replication factors, this promoted the idea that cells separate the nucleus into distinct domains with high transcriptional activity (termed transcription factories) and DNA synthesis (termed replication foci). This is further supported by a nascent RNA sequencing assay over the course of S-phase, which revealed a global anti-correlation of the replication timing and peak transcription of the gene, suggesting that cells have indeed evolved mechanisms to temporally and spatially separate the two processes during the cell cycle (Figure 1) [9].

Transcription complexes can pose a roadblock for replication by either moving in the same or the opposite direction, resulting in co-directional (CD) or head-on (HO) TRCs. The complex multi-subunit machineries assemble at distinct genomic regions named promoters and origins. Interestingly, the relative locations of these elements along eukaryotic chromosomes are not random, but efficiently firing replication origins tend to overlap with transcription start sites (TSSs) of highly transcribed genes, thereby promoting a global CD bias of transcription and replication [10,11]. Although this strategy can effectively minimize HO-TRCs, considered to be the more harmful and deleterious type of TRC in bacteria, yeast and higher eukaryotes [12,13,14,15], a recent study showed that a subset of TSSs remains under-replicated and relies on G2/M DNA synthesis (G-MiDS) to complete replication and prevent DNA damage in mitosis [16]. These G-MiDS hotspots remain persistently under-replicated in the S-phase due to RNAP complexes at the TSS that block replication fork progression. Thus, by deferring DNA synthesis into the following G2-phase, G-MiDS allows gap filling, thereby reassuring complete genome duplication (Figure 1, G2).

However, eukaryotic cells employ additional strategies to protect highly transcribed regions from HO collisions. For example, in the nucleolus, a polar replication fork barrier prevents HO collisions by blocking forks from progressing in the opposite direction to transcription at the highly transcribed ribosomal DNA gene clusters [17]. During the G1-phase, transcription may also have a more direct role in negotiating TRCs, as it was shown in vitro that intragenic MCM2-7 double-hexamer complexes can be repositioned to the transcription termination site by active “pushing” by RNA polymerase (Figure 1, G1) [18]. Whether this redistribution shows the same extent in the context of chromatin in vivo and whether this results in an inactivation of the origin are still open questions, but active transcription clearly has the potential to shape the landscape of replication initiation sites prior to S-phase entry [19,20].

Despite these multiple strategies that cells are equipped with to coordinate the two machineries and prevent TRCs, occasional encounters appear to be inevitable. The aim of this review is to discuss the consequences of such unscheduled and unresolved conflicts. Finally, we will provide an overview of the multiple pathways and partially redundant mechanisms that cells have put in place to process these conflicts, highlighting the importance of conflict resolution towards the overall goal of accurate genome duplication.

## 2. TRCs as a Potent Endogenous Source of DNA Damage and Genomic Instability

TRCs have been proposed as endogenous drivers of mutagenesis, recombination and other DNA alterations, thus representing potent threats to genomic integrity. However, studying the precise genomic consequences of these transient and likely short-lived events in vivo is a challenging task, particularly in higher eukaryotic genomes. Unlike in bacteria, where a single origin replicates each gene in a predictable orientation, eukaryotic chromosomes contain numerous origins that fire stochastically with variable efficiencies and timings [21,22]. Moreover, eukaryotic cells have excess origins that can complete DNA synthesis when other replication forks stall [23]. Thus, it is difficult to predict the location and orientation of a collision in eukaryotic genomes with absolute precision. In addition, both machineries can exist in different functional states. For example, RNAP complexes can be paused proximal to promoters, productively elongating along the gene body, or assume other configurations such as backtracking or the formation of R-loops. R-loops are three-stranded secondary DNA structures, where the nascent RNA strand rehybridizes with the complementary DNA template strand, resulting in an RNA/DNA hybrid plus displaced single-stranded DNA [24]. Although these structures form naturally during transcription and have been ascribed many physiological roles in cellular processes, the presence of R-loops in the context of TRCs is thought to stall transcription ahead of the replisome and thereby have a negative impact on genome stability (reviewed in [25]).

In summary, replisome collisions with different types of co-transcriptional obstacles—often present simultaneously at highly transcribed gene bodies—can display major differences in their outcome and severity of the TRC. In this chapter, we will discuss our current view on how TRCs can induce DNA damage and genomic instability, highlighting common results and differences among prokaryotic, yeast and mammalian reporter systems as well as at endogenous chromosomal loci.

### 2.1. Artificial Reporter and Plasmid Constructs as Model Loci for TRC Induction

#### 2.1.1. Prokaryotes

As mentioned above, prokaryotes typically harbor only one circular chromosome with a single origin of replication, resembling an optimally controlled system for studying different TRC orientations in a localized manner. By inserting inducible reporter or resistance genes into the prokaryotic genome in both orientations, such model loci can specifically detect differences between HO or CD collisions, as other confounding parameters such as gene sequence, transcription levels and the chromosomal location are identical for both constructs. For example, such inducible reporter genes (lacZ and luxABCDE) inserted on one arm of the *B. subtilis* chromosome demonstrated that HO but not CD conflicts provoke pervasive formation of R-loops. If not resolved by the activity of the RNase HIII enzyme, this can lead to complete replication blocks and elevated mutagenesis, suggesting that the instability derived from TRCs is, in part, mediated by the presence of RNA/DNA hybrids at the conflict site [14]. Interestingly, in vitro studies showed that replisomes bypass “naked” R-loops (in the absence of the transcription complex and other protein factors) in both orientations, suggesting that the genome-destabilizing effects of R-loops on replication fork stalling likely involve DNA–protein interactions such as the transcription complex or chromatin [26]. Consistently, it was shown that dCas9 protein-bound R-loop structures can arrest replisome progression, and bypass of this R-loop block relies on the monomeric Pif1 helicase in vitro [27].

Another study in *B. subtilis* took advantage of the fact that any complete loss-of-function mutation in the thymidylate synthetase reporter gene thyP3 can be selected using trimethoprim resistance, allowing for an in-depth analysis of the mutational spectra of HO- and CD-TRCs on the thyP3 gene. In general, the genetic consequences of TRCs in actively dividing bacteria can be categorized into insertions, deletions and base substitutions [28]. While insertions and deletions (indels) were observed for both CD and HO, their distribution across the gene body was reflective on where the first encounter of a replication fork entering a transcription unit occurred. Upon active transcription, CD collisions caused indels that are preferentially located in 5′ gene regions, whereas HO collision indels were distributed throughout the gene body and the 3′ region. Interestingly, HO collisions showed a specific increase in mutation rates at promoter sites that were mechanistically linked to deamination of specific promoter residues (Figure 2A) [28]. Collectively, both CD- and HO-TRCs can give rise to mutations and alter bacterial genomes. Nevertheless, HO conflicts exhibit a broader mutational spectrum with more severe consequences for genome integrity but also higher potential for evolutionary adaptation.

#### 2.1.2. Budding Yeast

The use of centromeric plasmids in budding yeast has also provided clear evidence that HO encounters have more severe effects on genome stability than in the CD orientation. By using constructs that contain two direct repeats of a 0.6 kb internal fragment of the LEU2 gene in either the HO or CD orientation to an early-firing replication origin, inducible transcription in the HO orientation could efficiently trigger the recombination of these repeats into a functional LEU2 gene. Importantly, this effect was much reduced in the CD orientation and was dependent on S-phase transcription, as a G1-phase-specific promoter had no effect on the frequency of recombination. These results strongly support a role for HO-TRCs in driving this recombination event (Figure 2B) [12]. Interestingly, RNase H overexpression in this system was shown to rescue the hyperrecombination phenotype in the HO construct, but it had no effect on CD encounters, also suggesting that RNA/DNA hybrids contribute to the instability of HO-TRCs [29]. Budding yeast has also provided an excellent model system to study engineered chromosomal CD- or HO-TRCs due to a detailed understanding of the location and activation timing of replication origins. For example, by inserting the tetracycline-regulated LYS2 reporter system in the opposite or same orientation to an efficient origin on chromosome III, the authors could show that transcription caused −2 frameshift mutations and complex deletions and insertions. Importantly, this occurred at higher rates in the HO orientation, indicating orientation-dependent effects on such mutation spectra (Figure 2B) [30]. Similarly, a LEU2 gene under a galactose-inducible promoter was inserted head-on to the early-firing ARS315 origin. Transcription-induced R-loop accumulation and phosphorylation of histone H2A as a marker of DNA damage was observed on the reporter gene, suggesting that HO transcription can also generate DNA breaks on this chromosomal TRC system (Figure 2B) [29]. Although such reporter constructs often select for particular types of DNA alterations such as recombination and may not capture all genetic insults of TRCs, these results underline the large potential of HO-TRCs to induce such genetic changes with a higher frequency than the corresponding CD encounters.

#### 2.1.3. Episomal System in Mammalian Cells

Unlike budding yeast, mammalian genomes do not seem to have well-defined, sequence-specific origins. Consequently, it has been difficult to identify the precise location of metazoan replication origins, and the common view is that cells have a large excess of replication origins that are used in each cell in a flexible and stochastic manner (reviewed in [31]). The resulting inability to predict the site and orientation of collisions represents a major challenge to study TRCs on mammalian genomes. To overcome this, a human cell-based plasmid system was developed that allows for controlled transcription and replication on episomal constructs in human cells. These constructs express different inducible transcription units that are prone to R-loop formation (e.g., the promoter region of the mAIRN gene) or control regions that are transcribed without R-loop formation (e.g., the ECFP gene). They also contain a single unidirectional replication origin (oriP/EBNA1) that recruits the endogenous replication machinery and is activated only once in the S-phase. By cloning the transcription units in both directions relative to the replication origin, R-loop-dependent and independent HO and CD collision events could be discriminated in human cells. Interestingly, DNA breaks were detected on both R-loop-forming constructs, but the orientation seems to determine how the events are processed, resulting in the activation of distinct DNA damage responses, namely, ATR kinase in R-loop HO and ATM kinase in R-loop CD constructs (Figure 2C). Consistent with results from bacteria and yeast cells, the analysis of R-loop levels on the mAIRN HO/CD constructs showed that RNA/DNA hybrid levels are elevated in the HO orientation, supporting a role for these structures in promoting TRC-induced genomic instability. Although the episomal system has the advantage that stalled forks cannot be rescued by converging forks from the other orientation—as would be the case on the chromosome—it is important to note that events on these short (~ 10 kb) plasmids may differ in, for example, topological aspects and not accurately model all the chromatin dynamics and endogenous chromosomal transactions in the genome. Thus, it will be important to evaluate results from the episomal system regarding, for example, the orientation-dependent DNA damage responses in a more physiological context. Based on the recent advances in Okazaki fragment sequencing (OK-Seq) to determine replication fork directionality (RFD) [10,32], one future direction could be to insert such mammalian reporter genes at predetermined genomic sites that exhibit near complete fork progression in one direction to allow preferential induction of HO or CD collisions in native mammalian chromatin.

### 2.2. TRC-Induced Genomic Instability at Endogenous Chromosomal Loci

#### 2.2.1. Highly Expressed Genes Challenge Replisome Progression

Intriguingly, bacterial model organisms, such as *B. subtilis*, show a very strong co-orientation bias of transcription and replication that is required to protect genomic integrity [32], whereas HO-oriented genes display increased mutation rates and potentially accelerate gene evolution [33]. Thus, the orientation of transcription and replication in CD appears to be an important characteristic of bacterial genome organization, safeguarding it against higher mutation rates and genomic instability. This bias is particularly important at highly transcribed and essential genes, as inversion of rRNA genes in the HO orientation disrupted replication and resulted in activation of the DNA damage response and cell death [34]. Nevertheless, CD encounters are not harmless and can also cause disruption of replication, as shown for the ribosomal RNA (rrn) genes in *B. subtilis* that accumulate the helicase loader proteins DnaB and DnaD required to restart replication forks at these chromosomal sites (Figure 3A) [35]. More recently, replication fork stalling has been directly observed in prokaryotes by expressing the replicative helicase DnaC as a GFP-tagged hybrid protein. The use of single-molecule fluorescence microscopy allowed tracking of replication fork movement in real time at the single molecule level and indicated that TRCs are much more pervasive, leading to frequent DNA replication discontinuities in up to 50% of ongoing replication forks (Figure 3A). Thus, at least in bacteria, transcriptionally impeded forks appear not to be rare, but rather represent one canonical state of replisome progression [36]. Importantly, every single fork must be rescued, and a failure to do so is a lethal event, explaining the essentiality of replication restart factors. Consequently, the accurate and faithful duplication of the genome relies on the DNA replication, repair and genetic recombination machinery working closely together, particularly at highly transcribed genes [37].

In yeast, it is less clear whether highly transcribed genes impede DNA replication fork progression. Highly transcribed RNAP II genes as well as RNAP III-transcribed transfer RNA (tRNA) genes were shown to slow fork progression and require Pif1 family DNA helicases to promote DNA replication past these particular sets of genes [38,39]. On the other hand, head-on collisions between tRNA transcription and replication appear to be under-represented in the presence of these helicases [40], and a more recent study on tRNA genes concluded that the nature of the block to replication may not be tRNA transcription per se, but rather the asymmetric binding of the essential TFIIIB transcription factor at tRNA promoters [41]. This result suggests that also other transcription-related factors such as initiation complexes can be a significant barrier to replisome progression.

#### 2.2.2. Common Fragile Sites and Early Replicating Fragile Sites

In more complex mammalian genomes, our understanding of TRC-induced mutational consequences remains limited, as TRCs might occur ubiquitously and presumably in a random manner throughout the S-phase, making their occurrence much less predictable. However, a role for TRCs has been suggested for a set of frequently instable genomic loci that present gaps or breaks on metaphase chromosomes after exposure to the DNA replication inhibitor aphidicolin [42,43]. These sites, termed common fragile sites (CFSs), are characterized by late replication timing, low density of replication origins and persistence of aberrant mitotic structures such as micronuclei or ultrafine anaphase bridges (UFBs) [44]. Correspondingly, the genomic positions of ongoing mitotic DNA synthesis (MiDAS) have recently been shown to encompass all known CFSs [45]. Interestingly, many CFSs harbor extremely large genes with long intronic sequences, and their extended transcription time could render them particularly susceptible to encountering a replication fork, inducing incomplete replication, chromosomal instability and acquisition of deletions [46,47,48]. As both transcription and replication processes have been proposed as essential contributors to the genomic instability at CFSs, TRCs could be an attractive molecular mechanism for their fragility. Nevertheless, their exact contribution remains an open question. At the CFS genes FHIT and WWOX, TRCs were demonstrated to be inevitable since their transcription frequently spans more than one cell cycle [5]. Additionally, replication stress-induced activation of downstream replication origins has been shown to give rise to TRCs. In this model, aberrantly activated origins are likely to interfere with the transcription and replication co-directionality and thus give rise to HO conflict-induced genomic instability [11]. These observations could explain how replication stress increases CFS fragility in a TRC-mediated manner. Nevertheless, genome-wide investigation of nascent transcription, replication origin positioning and fork directionality showed that the low replication origin density can explain CFS fragility, whereas TRC-induced replication delays do not [32]. Specifically, transcription inhibition in the S-phase, which mitigates TRC formation, was not able to rescue aphidicolin-induced CFS fragility on previously characterized CFS genes. Rather, transcription-mediated modulation of the replication initiation program, causing origin repositioning and thus origin paucity in CFS genes, was proposed to shape the tissue-dependent landscape of CFSs and genomic instability (Figure 3B). In addition, transcription was even demonstrated to prevent fragility at certain CFSs by advancing the replication timing to the earlier S-phase, providing more time to complete replication at these regions [49]. Despite differing views on the contribution of TRCs to CFSs, a co-existence of both proposed mechanisms is plausible. Some CFSs could be affected by TRCs, whereas for others, origin paucity is the crucial determinant of fragility. An in-depth evaluation of the effect of different types of replication stress on CFS breakage could also help to clarify the disparity between the currently proposed mechanisms. Ultimately, further investigations, especially on a genome-wide scale, will be required to understand the precise molecular mechanism underlying CFS fragility.

Apart from CFS, a distinct type of fragile sites was first described by Barlow et al. in 2013. These fragile genomic regions are characterized by their localization within highly expressed gene clusters, their close proximity to early replication origins and their enrichment for repetitive elements and CpG dinucleotides [50]. Due to their replication in the early S-phase, they are termed early replication fragile sites (ERFSs). Despite their distinct sequence as well as transcription and replication profiles, ERFSs’ stability is similar to CFSs’ by its dependency on the replication stress response kinase ATR [50]. Moreover, ERFSs’ fragility, in addition to spontaneous breakage, can be increased by hydroxyurea, ATR inhibition or deregulated expression of the c-Myc oncogene. Thus, the situation at ERFSs could favor transcription–replication interference as a possible explanation for their fragility [32,51]. Despite still lacking direct analysis of cell cycle-specific ERFS transcription patterns and TRC occurrence at these sites, oncogenic stress (see below), which likely alters replication dynamics at the highly transcribed ERFSs, as well as a transcription-mediated increase in ERFS fragility, supports the view that TRCs could at least contribute to ERFS instability [51,52].

#### 2.2.3. Global Perturbation of Transcription and Replication Programs in Eukaryotic Genomes

Although current methods to detect TRCs lack the molecular resolution to pinpoint the precise location of endogenous TRCs, many cellular systems have been developed to globally perturb the coordination of transcription and replication by the means of external stimuli or the use of specific inhibitors. In this way, TRCs are expected to be globally induced, allowing the study of their genomic outcomes. For example, it was shown that elevated osmotic stress can lead to the induction of osmo-responsive genes in the S-phase. As a result, the stress-activated protein kinase Hog1 is activated and phosphorylates the replication fork component Mrc1, thereby delaying fork progression and origin firing. Thus, this S-phase checkpoint permits eukaryotic cells to prevent TRC-mediated fork stalling and fork collapse [53]. Similarly, treatment of cells with hydroxyurea was shown to transcriptionally induce a subset of genes that are highly prone to DNA breaks as a result of destabilized replication forks encountering transcriptional complexes [54]. In human cells, short inhibition of the Cyclin-dependent protein kinase 9 (CDK9) leads to the induction of RNAP II stalling and the rapid colocalization of recombination repair factors in proximity to such potential TRC sites [55]. Other examples include the stimulation of breast cancer cells with estrogen that results in deregulated transcription and the formation of R-loops that induce replication-dependent DNA breaks [56] or the increase in torsional stress on the DNA by Topoisomerase I (TOP1) inhibition [41].

Oncogene activity is another emerging cause that can deregulate transcription and replication processes in numerous ways including origin licensing impairment, replication fork progression, premature S-phase entry, induction of re-replication and aberrant transcription regulation [57,58]. For example, overexpression of Cyclin E or MYC can induce firing of ectopic DNA replication origins within highly transcribed gene bodies. Under unperturbed conditions, transcription in the G1-phase would have sufficient time to clear such intragenic origins. However, oncogene overexpression leads to premature S-phase entry, and these oncogene-induced origins are prone to collapse as a result of TRCs [21]. Interestingly, the chromosomal breakpoints in this system significantly colocalized with chromosomal rearrangements observed in human cancer cells, providing a mechanistic link between TRCs and oncogene-induced genomic instability. Overexpression of cell division cycle 25A (CDC25A) was also shown to slow replication forks and induce fork reversal, inducing the DNA damage response in the S-phase [59], but a direct connection to TRCs has not been established yet in this system. Finally, recent studies demonstrated that overexpression of cyclin E or mutated HRAS in human fibroblasts is capable of inducing chromosome breakage which partially overlaps with classical APH-induced CFSs, leading to the establishment of oncogene-induced fragile sites [60]. Crucially, oncogene-stimulated transcription and replication dynamics are globally perturbed and highlight a potential mechanism by which transcription–replication interference, including TRC formation, could drive genomic instability in cancer [52,61]. While a different oncogene induction system showed a clear correlation between oncogene-derived genomic instability and CFSs [62,63], further investigations will be needed to clarify whether oncogenic TRCs directly occur at CFSs and induce their fragility. Altogether, these different techniques to perturb the coordination between transcription and replication show how delicate the equilibrium is between the two processes. A global perturbation of this balance can lead to widespread consequences for the genome, therefore linking TRCs with chromosome breakage, genome instability, replication fork collapse and other genomic catastrophes.

## 3. Crosstalk between the Chromatin Environment and TRCs

Transcription and replication machineries do not exert their essential functions on naked DNA, but these chromosomal transactions occur in the context of chromatin. Importantly, both processes—when seen on their own—already represent major challenges for cells to faithfully maintain or transmit epigenetic information to daughter cells. For example, replication forks dismantle nucleosomes on the parental strand to allow fork passage. The corresponding parental histones, with their specific post-translational modifications (PTMs), are deposited on the two daughter strands, along with newly synthesized histones (reviewed in [64]). Thus, chromatin has to be restored at all levels after fork passage, including the repositioning of nucleosomes [65], re-establishment of the parental histone PTM landscape on the new histone pool [66], recruitment of transcription factors and RNA polymerase to restore gene expression programs [67] and DNA looping and three-dimensional compartmentalization [68].

Similarly, nucleosomes represent major roadblocks for RNAP II transcription and need to be removed to allow RNAP complexes access to the template DNA. Recent cryo-EM studies showed that nucleosomes induce two major pauses of RNAP II elongation complexes shortly after 15 bp and 45 bp of nucleosome entry, which can only be overcome by the cooperative action of the transcription elongation factors Elf1 and Spt4/5 [69,70]. To maintain the epigenetic states in genes, nucleosomes need to be reassembled in the wake of RNAP passage. The best-studied histone chaperones that fulfill this important task are FACT (facilitates chromatin transcription) and SPT6 (Suppressor of Ty6), which recycle histones together with their local PTM landscape co-transcriptionally [71,72] and therefore prevent scattering of individual histones on multiple gene bodies or different positions along the same gene. Together, a clear prediction of these transcription- and replication-dependent chromatin maintenance pathways is that impediments to replication fork progression and transcription blockage can both—on their own—trigger local changes in the chromatin structure. As TRCs describe the simultaneous interplay of both machineries on the same chromatin template, it appears highly likely that such conflicts are a potent source of spontaneous chromatin changes arising at individual loci. Such a conflict-mediated chromatin “scar” may have particularly important consequences at essential or key disease or developmental genes that can lead to dysregulation of the transcriptional activity and rewire the gene expression programs of cells.

### 3.1. Histone PTMs as Regulators of Replication Fork Speed

Multiple histone marks are associated with different states of gene transcription and form co-transcriptionally at distinct steps in the RNAP II transcription cycle. Among them are the H3K4me3 mark for initiation, H3K9ac for pause release, H3K36me3 for elongation and H3K9me2 for termination [73]. Depending on the preferred location of the collision site, one can hypothesize that conflict-induced RNAP II stalling can delay, reduce or inhibit the proper deposition of these marks as part of the transcription cycle and thereby provide a mechanism to change the transcriptional output of the gene. In fact, methylation of histone 3 lysine 4 (H3K4) was recently connected with TRCs. H3K4me3 is widely accepted as an activating histone mark at TSSs, but actively transcribed genes display a 5′ to 3′ gradient of H3K4me2 and H3K4me1 marks in the gene body. Using a system for inducing TRCs, the authors found that H3K4 methylated regions serve as “speed bumps” for replication forks, slowing down fork progression in the face of strong transcription and thereby preventing TRC-induced genomic instability [74]. Furthermore, N-terminal H3 acetylation mediated by the acetyltransferase Rtt109 was shown to slow the replication fork speed in a wave of 3–5 kb ahead of the fork, which could also contribute to proper nucleosome replacement, thereby promoting genome stability [75].

Although the precise molecular mechanism(s) of how such epigenetic marks can “crosstalk” with the replisome is (are) currently missing, these examples illustrate that chromatin replication can be affected by a specific transcription-associated histone mark. In the future, it will be interesting to test whether other activating marks also impact and fine-tune fork progression and replication fidelity at TRC-prone loci, including the SETD2 histone methyltransferase-deposited H3K36me3 mark that is enriched for genes with high RNAP II and R-loop density [76].

### 3.2. Chromatin as an Insulator against Transcription-Induced DNA Damage and Replication Stress

Chromatin constitutes a dynamic nucleoprotein complex with a primary function to compact the genome and thereby protect the DNA from genomic insults and promiscuous activities such as cryptic transcription. To facilitate or restrict access to the DNA, chromatin remodelers, histone chaperones or other specific factors are recruited to regulatory regions of the genome. Thus, any perturbation in the chromatin structure or chromatin regulatory factors can abolish this insulating function of chromatin and impact both transcription and replication dynamics. For example, changing the histone/DNA ratio by deletion of multiple histone H1 copies in mouse embryonic fibroblast (MEF) cells caused massive replication stress and DNA damage signaling that depended on active transcription, suggesting a potential role in TRC occurrence. Interestingly, altering chromatin dynamics by depletion of the HMG box containing protein HMGB1 allowed faster fork progression in the same cells but without changing the replication landscape or causing fork instability, implying that cells can tolerate certain chromatin perturbations even though they lead to variable speeds of replication and transcription complexes [77].

An incomplete or suboptimal chromatin structure may also be a causal reason for the formation of co-transcriptional R-loops which can, in turn, lead to replication fork impairment and TRCs. This is evidenced by the histone chaperone FACT and the SIN3A histone deacetylase complexes that were shown to prevent R-loop-dependent TRCs in yeast and human cells [78,79]. Interestingly, histones appear to play a direct role in transmitting R-loop-induced DNA damage as certain histone H3 and H4 mutants suppress the genomic instability phenotypes despite strong R-loop accumulation in these cells [80]. Thus, chromatin may significantly contribute and amplify the cellular response to R-loop-mediated TRCs. However, the exact relationship and sequence of events in the formation of R-loop structures, TRCs and the recruitment of chromatin factors are not well established. Several studies in bacteria and human cells have indicated that co-transcriptional R-loops can accumulate specifically at HO collision sites [13,14,15], although it was shown in yeast that co-transcriptional R-loops can form initially at all coding regions, independently of fork orientation [29,81]. Thus, the relative levels and position where such R-loops form in relation to the replication fork and how they are interfering with the replication process are still open questions.

The non-B-form helical structure of RNA/DNA hybrids is unlikely to be able to wrap around nucleosomes, raising the possibility that R-loops can create local nucleosome-depleted regions. Consistent with this notion, R-loops were shown to prevent nucleosome formation at regulatory regions of the vimentin gene in human colon cancer cells [82]. On the other hand, R-loops were previously indicated to induce chromatin compaction and accumulate histone marks of condensed chromatin including H3S10 phosphorylation and H3K9 dimethylation [83,84,85,86,87]. This apparent contradiction could be explained by the formation of larger chromatin domains that encompass both types of chromatin. In fact, RNA/DNA hybrid interactome studies revealed over 400 specific protein interactors including helicases, DNA repair, and chromatin factors [88], suggesting that these structures are likely adopted in cells and recruit different factors in a context- and location-specific manner (see also Chapter 4 below). Not surprisingly, an increasing number of reports connect now different chromatin factors with R-loop homeostasis, including the ATRX chromatin remodeling complex that was demonstrated to suppress R-loops at telomeric repeats [89] and the Tip60-p400 histone acetyltransferase complex that is tightly associated with genes harboring promoter-proximal R-loops. The presence of R-loops at these loci was also shown to decrease the occupancy of the PRC2 histone methyltransferase, a key regulator of development and chromatin structure [90]. Together, these examples highlight the essential functions of chromatin to protect the genome from different types of transcription-induced fork impediments.

### 3.3. Chromatin Compaction and Torsional Stress

As both transcription and replication generate positive supercoiling ahead, torsional stress of the DNA fragment between the two complexes is considered a major reason for the detrimental outcomes of HO collisions. In *B. subtilis*, the removal of this excess topological stress by gyrase is critical for TRC resolution [15]. Similarly, human cells depleted in TOP1 show an accumulation of R-loops, markers of replication stress and DNA breaks at transcription termination sites, which are preferentially replicated in the HO orientation relative to transcription [91], suggesting that TRCs can occur at transcription termination sites (TTS) of highly expressed genes.

Importantly, positive torsion has been described to destabilize single nucleosomes in the context of elongating RNAP complexes [92], although recent single-molecule studies indicate that nucleosomal arrays show a larger degree of elasticity and can absorb an excessive positive twist. Interestingly, a large number of positive turns refolded the chromatin fiber into a compacted state by creating new inter-nucleosomal stacking interactions [93]. However, the additive positive supercoiling of both machineries at HO conflict sites may exceed the capacity of topoisomerases to neutralize the torsional stress in vivo, thereby leading to the destabilization of the chromatin fiber. The precise molecular consequences for the nucleosomal landscape as well as the size of this DNA region “trapped” between the two machineries are currently not clear, but the build-up of torque in the DNA helix could impair efficient histone disassembly and reassembly between both machineries. The importance of chromatin remodeling complexes to maintain the chromatin topology in this process is also highlighted by the recent finding that the human BAF (SWI/SNF) complex controls R-loop levels associated with replication stress, DNA breaks and TRCs. In particular, the DNA-binding ARID1A subunit appears to be responsible for targeting TOP2A to genomic regions prone to TRCs [94]. The contribution of the SWI/SNF complex was independently confirmed by the Aguilera lab, showing that the main ATPase BRG1 colocalizes with R-loops and helps in resolving R-loop-mediated TRCs [95]. These findings unveil how defects in chromatin structure—mediated by R-loops, the loss of histones or chromatin remodelers—can impact the coordination of transcription and replication, leading to replication stress that could, in turn, enhance cellular aging and disease states such as cancer.

## 4. Mechanisms to Resolve a TRC

To avoid the deleterious consequences of TRCs, cells need to rapidly detect and resolve the conflict. TRCs are composed of three main elements, the replisome, the RNAP and the underlying DNA template. In general, the following possibilities are conceivable to resolve a TRC and restore a functional chromatin template. Upon conflict, the replisome can skip the RNAP complex by repriming downstream of it. In the case of a CD-TRC, the newly synthesized RNA can be used as a primer for replication restart. Alternatively, the RNAP can be degraded or expulsed from the chromatin to allow resumption of DNA replication. Another possibility is to transiently cleave and re-ligate the replication fork to allow the RNAP to resume transcription and move past the replication fork. For all these pathways, the replisome can undergo an intermediate step named fork reversal that stabilizes and protects the fork while allowing the cell more time to resolve the conflict.

When the conflict is not properly resolved, it can lead to the formation of a double-strand break (DSB), triggering DNA repair pathways such as break-induced replication (BIR) [96]. Lastly, if the cell is unable to resolve or skip the conflict, the replisome can be disassembled, and replication is completed by an upstream converging replication fork or by activation of an upstream dormant origin. This chapter will focus on the most recent discoveries regarding the different cellular mechanisms that are put in place at sites of conflicts to regulate their processing, thereby allowing complete genome duplication.

### 4.1. RNAP Skipping and Repriming

Once confronted with a co-directional stalled RNAP, the replisome can displace the RNAP and use the newly synthesized RNA molecule as a primer to reinitiate replication, facilitating TRC skipping (Figure 4A) [97]. In vitro studies using reconstituted *E. coli* replisomes showed that DNA replication was only transiently interrupted by a CD RNAP, whereas the majority of RNAPs could be displaced by the replicative helicase [26,95,96]. Although these highly controlled in vitro reactions are well designed to provide insights into the kinetics of conflicts, the situation in vivo is likely more complicated. With multiple RNAPs transcribing the same gene simultaneously, intrinsically different genomic sequence contexts and additional protein factors not present in these minimal systems, TRC studies are complex and multifaceted. More recently, a series of elegantly designed in vitro templates was used to investigate how the *E. coli* replisome deals with more complex RNAP arrays as well as replication-R-loop collisions. Consistent with previous results [27,28,29,30], co-directional RNAP complexes imposed only transient blocks, whereas the HO orientation led to severe fork stalling, particularly when challenged with an array of multiple RNAPs. Altogether, these results provide insights into the robust enzymatic activities present at replisomes to unwind secondary structures and to displace both HO and CD RNAPs, allowing RNA takeover in the case of CD-TRCs (Figure 4B).

Another possibility for the cell is to skip the block and resume DNA replication by synthesizing a new Okazaki fragment downstream of the RNAP complex (Figure 4A) [96,98]. This appears to be a simple and effective solution that takes advantage of the discontinuity of the lagging strand replication machinery to overcome obstacles on this DNA strand. However, for the leading strand, skipping and repriming imply the interruption and uncoupling of continuous DNA synthesis at the replication fork, thereby leaving stretches of ssDNA that will have to be filled by post-replicative repair pathways [96,98]. Repriming of the leading strand is executed by the intrinsic, although inefficient, ability of specialized DNA polymerases in both bacterial and eukaryotic cells [96,98]. In human cells, this activity is achieved by the dual polymerase and primase activities of the PRIMPOL protein (Figure 4A) [99,100]. Cells lacking PRIMPOL show an increased sensitivity to replication stress, suggesting its importance in leading-strand lesion skipping and repriming [101,102]. Furthermore, PRIMPOL does not travel with the replication fork, indicating that this factor can specifically recognize and is actively recruited to stalled replication forks [103]. It was also shown to reprime downstream of non-B-DNA co-transcriptional structures such as G4 and R-loops, supporting the idea that it can promote TRC skipping and repriming [96,104].

It is important to underline that it is still unclear whether the replisome can effectively traverse a stalled RNAP complex per se. In vitro reconstitution of conflicts using the T4 replisome shows that the replisome is able to pass the bound RNAP without dissociating it from DNA [105]. On the contrary, the *E. coli* replisome seems to displace both CD and HO RNAPs and reprime past the obstacle. In eukaryotic cells, bulky protein–DNA adducts and interstrand cross-links (ICLs) were considered as absolute blocks for the replicative helicase, and as such, bulky lesions cannot be accommodated by its central channel. Surprisingly, replication restart past an ICL was demonstrated, suggesting that the helicase could adopt an open-ring conformation to translocate past ICLs [106,107]. Lastly, the replicative helicase was also shown to bypass covalent DNA–protein cross-links (DPCs) on both the leading and lagging strands when assisted by accessory helicases such as RTEL1, allowing the re-engagement of the replicative helicase downstream of the DPC [108]. Nonetheless, whether such mechanisms allowing DPC bypass are used for the bypass of larger complexes such as RNAP is an open question.

### 4.2. TRC-Induced Removal of RNA Polymerases

Another option to resume DNA replication is to remove the RNAP block, for which multiple pathways exist (Figure 4C). One mechanism to evict RNAP complexes is mediated by accessory replicative helicases that can remove DNA-associated protein complexes in front of the replisome. In *E. coli*, the helicases Rep, UrvD and DinG display this function both in vitro and in vivo. In yeast, the helicase Rrm3 facilitates replication through protein–DNA complexes and heavily transcribed regions [109,110]. RNAP complexes are also equipped with specific transcription factors, such as GreA/B in *E. coli*, that promote RNAP removal by nascent RNA cleavage upon TRCs [111]. Additionally, transcription factors that allow productive RNAP elongation or termination ensure proper RNAP movement on the chromatin and limit RNAP stalling [111]. In eukaryotes, reducing RNAP stalling and maintaining productive transcription rates are expected to alleviate replication stress in a similar manner. This can be exemplified by the effect of RNAP II mutants in yeast with increased chromatin retention [112]. In these mutants, the replication stress induced by the inability to properly evict RNAP II from the chromatin results in more origin firing, presumably to rescue replication forks that have been stalled by RNAP complexes [112]. More recently, the transcriptional co-activator BRD4 was also shown to promote replication fork movement through actively transcribed genes, and its down-regulation leads to increased R-loops, replication fork slowing and DNA damage [113,114].

RNAP stalling at sites of DNA lesions is well established in the context of transcription-coupled nucleotide excision repair (TC-NER). The first step of this DNA repair pathway is the targeted removal of the stalled RNAP by the proteasome, giving access to the underlying DNA lesion [115]. This mechanism works in a two-step process and involves, first, the Rsp5 (NEDD4) protein, which is responsible for the monoubiquitylation of RNAP II [116,117]. Second, the Elongin-Cullin complex (Elongin ABC-Cullin5), together with Def1, is recruited to monoubiquitylated RNAP II and triggers its polyubiquitylation, allowing the Cdc48-dependent targeting of RNAP II to the proteasome [116,117]. Although the TC-NER pathway might not be actively recruited at sites of TRCs in particular, it represents an important cellular mechanism to remove stalled RNAPs and therefore prevent TRCs by clearing the path for replisome passage. Interestingly, human fibroblasts display an S-phase progression-dependent apoptosis upon UV irradiation, suggesting that damage–stalled RNAPs can strongly impair DNA replication [118]. Thus, removing stalled RNAPs as a replication block can prevent TRCs and allow unrestrained fork progression in S-phase.

Interestingly, TRCs occurring in different genomic contexts appear to rely on specific pathways for RNAP removal. For example, the centromeric repeats depend on the RNAi pathway for RNAP eviction in *S. pombe*, as in the absence, stalled replication forks accumulate and rely on the homologous recombination pathway to resume DNA replication [119]. On the other hand, the Dicer protein Drc1 regulates transcription termination and RNAP II release in an RNAi-independent manner [120]. Indeed, drc1∆ cells show accumulation of RNAP II at sites enriched for replication fork pausing, including actively transcribed genes, the rDNA locus and tRNA genes [120]. This suggests an additional RNAi-independent role of Drc1 in removing RNAP II from the chromatin at TRC sites and shows the high diversity of RNAP removal pathways.

There is also evidence that the proximity between a replication fork and an RNAP complex can trigger post-translational modifications to regulate the RNAP removal from chromatin and its subsequent proteasomal degradation to resolve TRCs in yeast. The yeast ATR kinase homologue Mec1 is responsible for the phosphorylation of many effectors, including the nucleosome remodeling complex INO80, which travels with the replication fork, as well as the PAF1 complex, which travels with RNAP II and promotes transcription elongation and 3′ end processing of mRNAs and snoRNAs. In particular, it was demonstrated that both INO80 and PAF1 complex subunits are Mec1 substrates, and all three complexes show physical interactions upon replication stress [121]. Given that Mec1 accumulates at stalled replication forks and that cells lacking either Mec1, a functional PAF1 or INO80 complex all show a similar inability to remove RNAP II from chromatin under replication stress, these data suggest a model where these proteins participate together in regulating RNAP eviction specifically at TRC sites [121]. Interestingly, deletion of subunits of the PAF1 complex or of the INO80 complex has an additive effect on the sensitivity to replication stress of the mec1-100 mutation, implying that these complexes might also work separately to assure genome-wide protection against TRCs [121].

Regulation of the phosphorylation state of the RNAP itself also plays a role in TRC resolution. Recently, it was demonstrated that the protein phosphatase 1 (PP1), nuclear targeting subunit (PNUTS) and its binding partner WDR82 reduce replication stress by promoting RNAP II removal from the chromatin and its degradation [122]. PNUTS is responsible for directing PP1-mediated RNAP II CTD Ser5 dephosphorylation [123,124]. This promotes the RNAP II degradation and thus suppresses TRCs [122]. Indeed, depletion of PNUTS and/or WDR82 by siRNA causes a higher RNAP II retention on chromatin, as measured by ChIP and FRAP experiments, in addition to slowing down replication rates, reducing the recovery of replication forks after stalling and promoting ATR activation [122]. Interestingly, these effects depend on CDC73, a member of the PAF1 complex, as its co-depletion partially restores the observed replication defects, suggesting either different roles of the PAF1 complex through evolution or potentially a dual role of the PAF1 complex [122].

These diverse pathways regulating RNAP eviction show the importance to remove stalled RNAP complexes quickly and efficiently from chromatin, thereby preventing and/or resolving TRCs. This apparent redundancy of multiple pathways may represent a safety mechanism if one pathway is impaired or may be used in different genomic contexts or functional states of the complexes, for which different proteins might be more suited than others to process the conflict.

### 4.3. Processing R-Loops at Conflict Sites

As mentioned above, R-loops can, outside of their physiological roles in cells, exacerbate collisions between transcription and replication [25]. Hence, many mechanisms have evolved to reduce R-loop formation such as the co-transcriptional coupling of RNA splicing, maturation and export processes [25,125,126,127,128,129,130]. Proper maintenance of DNA compaction and regulation of DNA topology also serves the purpose of R-loop inhibition [76,78,80,91,131,132,133]. The most well-studied factor playing a role in the removal of R-loops is the ribonuclease RNase H, responsible for the degradation of the RNA part of an RNA/DNA hybrid [25]. Interestingly, the endonucleases XPG and XPF have been shown to recognize and cut R-loops as a non-canonical substrate that can be processed by the TC-NER pathway [134]. Alternatively, R-loops can be removed by the action of ATP-dependent RNA/DNA helicases such as Senataxin, FANCM, BLM and AQR [135,136,137,138,139]. In particular, both yeast Sen1 and Senataxin in human cells have been well described to preserve the integrity of replication forks encountering transcription complexes by removing RNA/DNA hybrids. For example, in the absence of Sen1, replication is strongly blocked by HO-TRCs [137], and the activation of dormant origins is required to complete replication [140]. Mechanistically, it was shown that yeast Sen1 binds and travels with the replisome directly via an interaction of its N-terminal domain with Ctf4 and Mrc1, thereby promoting fork progression and chromosome stability across R-loop-prone loci [141]. Over recent years, many additional DEAD-box RNA helicases have been added to this list of RNA/DNA helicases that can process R-loops. These include DHX9, DDX1, DDX3, DDX5, DDX19, DDX21, DDX23, DDX39B, DDX43, DDX47 and DDX56 [103,142,143,144,145,146,147,148,149,150,151,152]. TonEBP was also recently identified as an R-loop sensor and shown to recruit the RNA methyltransferase-like 3 (MTLL3) specifically to N6-methyladenosine (m6A) methylated R-loops [153,154]. This R-loop modification seems to play an important role in R-loop sensing and processing as it recruits RNase H for the degradation of the RNA [154].

Additionally, TRC-specific R-loop processing regulatory pathways have been identified. It was shown that FANCD2, a protein implicated in DNA ICL repair, is needed for the removal of RNA/DNA hybrids at CFSs [155]. This effect has been attributed to a function of FANCD2 in an early cellular response to CD-TRCs, together with the BLM helicase and the homologous recombination protein BRCA2 [55]. In consequence, disruption of this pathway leads to DNA damage [55]. The precise role of these proteins at TRCs is still unclear, but the recruitment of the RNA/DNA helicase BLM and the interaction of FANCD2 with DDX47 point towards a role in R-loop processing at TRCs [55,147]. Interestingly, the recruitment of these proteins to TRCs is ATR-independent [55]. Considering the role of ATR signaling in head-on TRC signaling [13] and in many TRC-resolving pathways [121], this might suggest that these proteins are recruited at co-directional TRCs.

ATAD5 was also demonstrated to play a role in the resolution of TRCs, particularly in the resolution of R-loops formed at TRCs [156]. ATAD5 is a PCNA unloader that interacts with the replication factor C (RFC)-like complex (RLC) to promote PCNA unloading during replication termination [157,158]. It also recruits the ubiquitin-specific protease 1 (USP1)/USP1-associated factor1 (USF1)-deubiquitinating enzyme complex to remove PCNA monoubiquitylation and regulate translesion synthesis (TLS) [158]. In addition, ATAD5 promotes replication restart after replication stress by removing PCNA from the fork and promoting RAD51 recruitment [159]. Through these functions, ATAD5 plays a central role in normal replication progression and in adapting to replication stress. Recently, it was also shown that ATAD5 depletion induces TRCs and leads to replication fork slowing [156]. ATAD5 presence at the replication fork then allows the recruitment of DEAD-box RNA helicases to PCNA to remove R-loops and facilitate replication fork progression [156]. In addition, it was proposed that ATAD5 promotes PCNA unloading at TRC sites to avoid the accumulation of PCNA behind the fork, which could trigger additional conflicts downstream of it [156].

Together, the presence of multiple and redundant R-loop processing pathways indicates that R-loops are a crucial part of the replication block, and that cells need to carefully negotiate their many physiological roles from their harmful consequences at TRC sites.

### 4.4. Other Modes of RNAP Skipping at TRC Sites

In the presence of a replication block that does not allow skipping and repriming or removal and resumption of DNA replication, the replication fork can be stabilized by a process named fork reversal (Figure 4D). Fork reversal is a complex, ATP-dependent remodeling process leading to the annealing of the two nascent strands together to form a “chicken foot” structure (reviewed in [160]). Fork reversal stabilizes and protects the stalled replication fork from nucleases, thereby allowing the cell more time to remove the replication block. More recently, it was shown that replication forks can undergo a cleavage and re-ligation cycle, allowing the RNAP to travel past the replication fork and DNA replication restart (Figure 4E) [161]. This mechanism depends on the RECQ1 and RECQ5 helicases, the SLX4 scaffold protein, the MUS81/EME1 endonuclease, RAD52, the DNA ligase IV, the DNA polymerase δ subunit POLD3 and the transcription elongation factor ELL, in a multistep process. Reversed forks at head-on TRC sites are initially remodeled back to the standard three-way fork configuration by RECQ1. Then, RECQ5 removes RAD51 from the stalled fork to inhibit subsequent fork reversal and to generate a substrate for MUS81/EME1-mediated cleavage of the fork. This enables the removal of torsional stress and allows the stalled RNAP complex to continue transcription past the site of conflict. Indeed, inhibition of transcription after the generation of a TRC does not permit replication restart as the RNAP complex is still present and blocks replication. Finally, RAD52 and DNA ligase IV act together to catalyze fork re-ligation and, therefore, facilitate replication restart [161].

Interestingly, RAD52 and LIG4, but not DNA replication, are needed for transcription recovery, suggesting that transcription needs the replication fork to be restored in its original configuration [161]. This mechanism is reminiscent of mitotic DNA synthesis (MiDAS), a mechanism by which fragile sites are replicated in mitosis after replication stress [45]. Interestingly, fork cleavage/re-ligation is the only mechanism uncovered so far that gives priority to the transcribing RNAP over the replisome, allowing it to complete its transcription instead of evicting it from the chromatin. This comes with the added risk of DSB formation, potentially increasing the risks of chromosomal rearrangements and genomic instability. This might represent the preferred mechanism of resolving TRCs at long genes where completion of the transcription cycle may be more valuable for the cell. Thus, an interesting avenue would be to determine if such loci show an enrichment for factors of this TRC-resolving pathway.

## 5. Conclusions

TRCs can be observed in consequence to numerous cellular stresses that affect different DNA transactions such as DNA replication, transcription, RNA splicing and DNA damage repair. It is therefore not surprising that TRCs are proposed to be crucial drivers of genomic instability, a hallmark of many human diseases. Although an increasing number of studies point towards an elevated TRC burden in many pathological cell states, it remains unclear to which extent they play a direct role in their establishment.

One interesting hypothesis regarding the potential outcome of TRCs is that they could perturb the correct transmission of histone marks and the correct reassembly of nucleosomes during both transcription and replication. Indeed, these processes are coordinated, and it is still unclear if and how the cell is able to properly maintain its epigenetic state after chromosome duplication and cell division. Any perturbation in histone mark deposition during replication could have long-lasting effects on transcription programs and genome organization. As such, it remains to be determined whether and to what extent TRCs have the potential to alter the epigenetic landscape of cells, which, in turn, might result in overall pathological transformations of cells. Additionally, TRC-mediated chromatin alterations could also correspond to an endogenous and physiological way to introduce subtle but important cell-to-cell variabilities in transcription programs. Such heterogeneity has been proposed to be of importance to allow cellular differentiation and to facilitate cell fate decisions. It would be an exciting avenue to investigate if cells have evolved in a way to use these “molecular accidents” in a physiological context.

In this review, we introduced the variety of genomic contexts that can induce TRCs and many redundant mechanisms that cells employ to process these conflicts. This redundancy could have several implications. First, it could represent a safeguard mechanism to avoid, for example, ERFS and CFS instability and breakage, as many of these pathways are impaired in tumors. Second, it may possibly reflect the variety of genomic contexts and functional states of the machineries that need to be resolved. It is conceivable, for example, that some mechanisms are more prevalent on conflicts involving an initiating RNAP complex and others on conflicts with an elongating RNAP complex. The different states of the RNAP, the genomic and nuclear localization, the epigenetic state of the chromatin and the S-phase timing could all represent factors that affect the severity of a TRC and the need for a particular resolution pathway. It will be interesting to investigate if some of these pathways show a preference towards specific “TRC substrates”.

A crucial roadblock in the investigation of the roles of TRCs lies within the lack of technical tools to study them appropriately. The ubiquitous nature of both machineries constituting TRCs renders their study particularly difficult. TRCs are often delineated from an increase in replication fork stalling or other indirect readouts. As these events are not exclusive to TRCs, it is difficult to prove their causal relationship. This problem is also pertinent regarding the different tools used to induce TRCs to study their effects. Indeed, manipulating the replication timing, transcription elongation, R-loop processing or other processes known to induce TRCs seems insufficient to imply a predominant role of TRCs because the manipulation of these processes potentially has pleiotropic effects. In this context, new tools allowing specific identification and manipulation of TRCs would be desired. As many of the TRC sensing and resolution mechanisms use proteins that normally travel either with the replication fork or with the RNAP complex, it would be a great advancement to the field to identify TRC-specific factors. Finally, technologies enabling genome-wide mapping of TRCs and subsequent correlation with observed mutation patterns would significantly improve our understanding of TRC occurrence, mutational consequences and disease contribution. With such methods in hand, one could start to investigate the link between TRCs, chromatin and associated genetic and epigenetic instabilities. The verification of a direct role of TRCs in diseases and in cellular plasticity would shed new light on the delicate cellular balance to coordinate these two genomic processes.

## Figures and Tables

**Figure 1 life-11-00637-f001:**
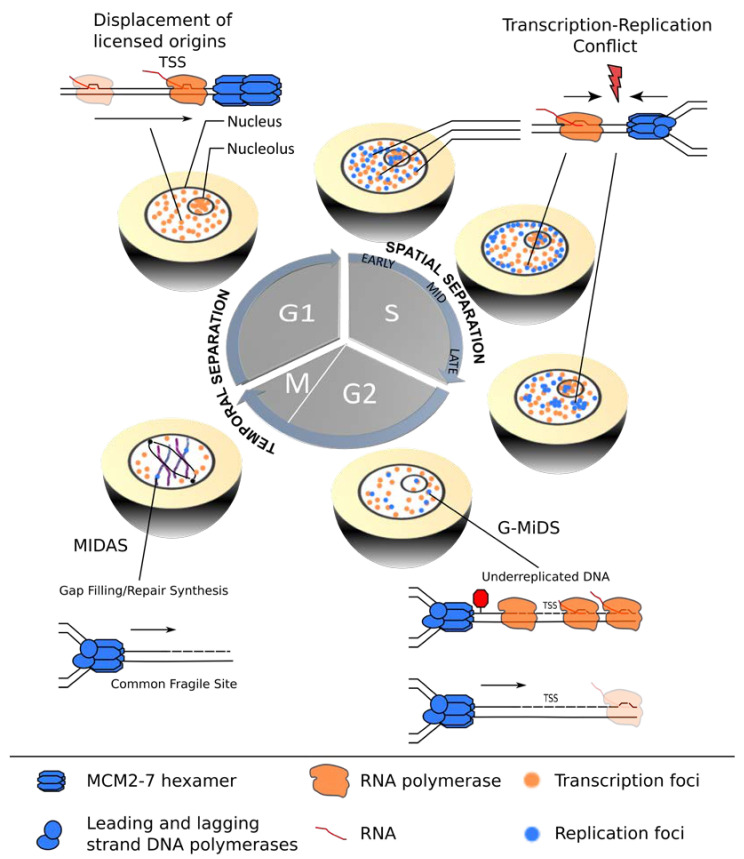
Coordination of transcription and replication over the cell cycle. In G1-phase, transcription (orange) is temporally separated from replication (blue), allowing for redistribution of replication origins from actively transcribed gene bodies, thereby assuring improved spatial coordination of the two machineries in the subsequent S-phase. TRCs are most likely to occur in early S-phase cells when significant overlap between active transcription and replication sites exists. Mid- and late S-phase stage cells show improved segregation of the nuclear transcription (orange) and replication (blue) foci. However, certain TSSs remain under-replicated and require G2/M DNA synthesis (G/MiDS) to complete genome replication. Finally, difficult-to-replicate regions, such as common fragile sites (CFS), rely on mitotic DNA synthesis (MIDAS) to complete genome duplication and ensure genomic stability during mitosis.

**Figure 2 life-11-00637-f002:**
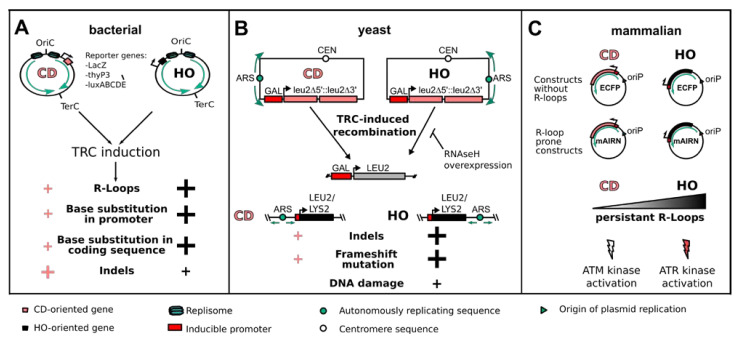
Consequences of TRCs on genome stability as studied by reporter systems. (**A**) In bacterial cells, the use of reporter genes such as LacZ, thyP3 or luxABCDE, in the CD or HO orientation can be used to induce TRCs and study their mutational outcome. Even though HO-oriented conflicts are more deleterious, both orientations induce R-loops and mutation of gene promoter and coding sequences. Additionally, insertions and deletions (indels) are more abundant at CD-TRCs, preferentially at the promoter region. (**B**) In the model organism *S. cerevisiae*, LEU2 reporter constructs were engineered in the CD or HO orientation to study the recombinogenic outcome of TRCs. In addition, the integration of an inducible LEU2 or LYS2 gene in the CD or HO orientation to a known replication origin allows studying the genetic outcome of TRCs in the chromosomal context. HO-oriented TRCs showed a stronger R-loop-dependent induction of recombination as well as a higher frequency of indels, frameshift mutations and DNA damage. (**C**) In mammalian cells, the cloning of either the mAIRN gene (R-loop prone) or of the ECFP gene (without R-loop formation) in a vector either in the CD or HO orientation relative to a viral unidirectional replication origin (oriP) allows discriminating between R-loop-prone and non-R-loop HO- or CD-TRCs. HO-oriented TRCs show persistent R-loop formation and activation of the ATR kinase, while CD-oriented TRCs show low levels of R-loop formation and signaling via the ATM kinase.

**Figure 3 life-11-00637-f003:**
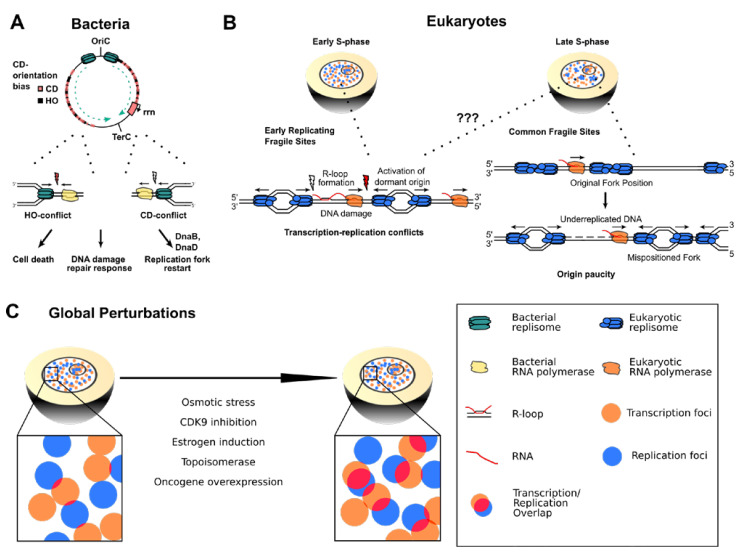
(**A**) In bacterial chromosomes, replication forks are frequently impeded by HO transcription and highly expressed CD transcription units such as the ribosomal RNA gene cluster (rrn). HO encounters typically lead to more severe consequences such as cell death and DNA damage repair response, but also CD encounters require replication restart factors to complete genome duplication. (**B**) Left panel: Arising TRCs at ERFS could lead to prolonged stalling of transcription and replication, favoring R-loop formation and DNA damage that can cause chromosome breakage and ERFS instability. Right panel: Transcribing RNA polymerases shift the position of replication origins on cellular DNA. This misplacement leads to origin paucity and long-distance traveling replication forks in the late S-phase that cause under-replicated DNA regions responsible for CFS fragility. (**C**) Global perturbations of replication and transcription dynamics can lead to imbalance of the two processes and globally higher transcription–replication interference.

**Figure 4 life-11-00637-f004:**
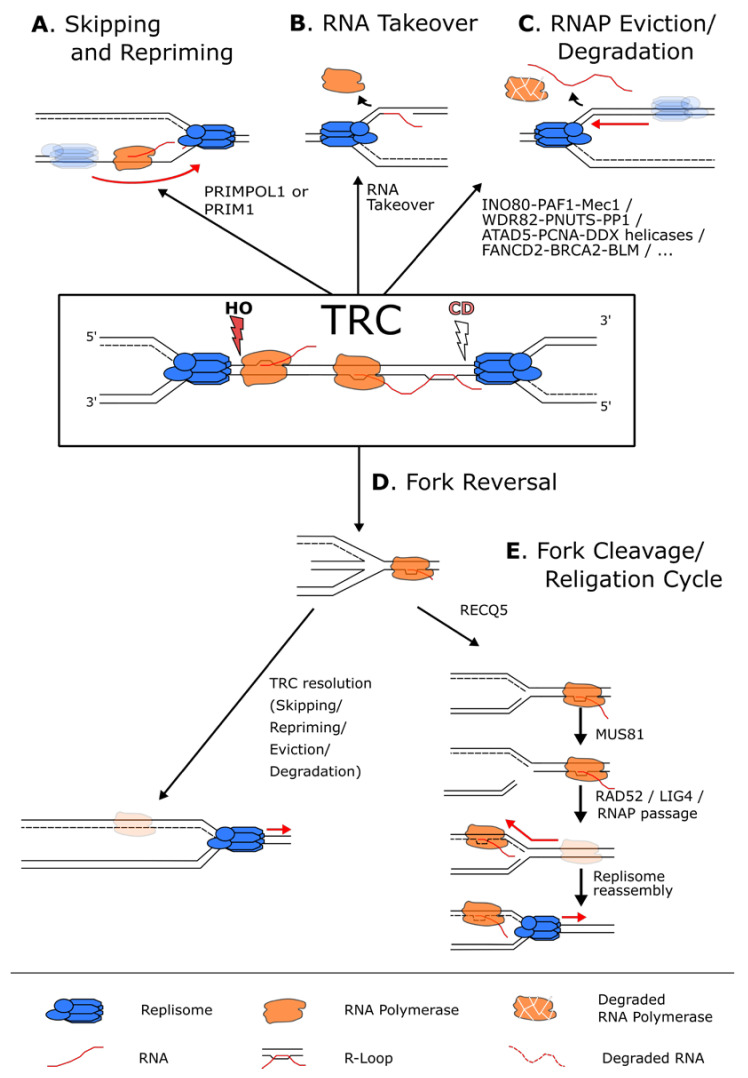
Pathways used to resolve a transcription–replication conflict. (**A**) When faced with a transcriptional block, the replisome (in blue) can skip the RNAP (in orange) and reprime downstream using the PRIM1 primase when the block is on the lagging strand, or the PRIMPOL1 polymerase when the block is on the leading strand. (**B**) In case of a CD conflict, the replisome can also displace the RNAP and use the hybridized RNA as a primer to reinitiate replication. (**C**) Numerous pathways exist to simply remove and, in some cases, degrade the RNAP and the potential associated R-loops from the chromatin to allow continuous DNA synthesis. (**D**) In case of persistent RNAP complexes, the replication fork can undergo fork reversal to stabilize the fork and give time to resolve the conflict by the above-mentioned mechanisms. (**E**) The reversed fork can also undergo a cycle of fork cleavage and re-ligation. This mechanism uses RECQ5 to inhibit fork reversal and the endonuclease MUS81 to cleave the fork. This relieves torsional stress, and the fork can then be re-ligated by RAD52 and LIG4 that in turn allows the resumption of transcription and the removal of the replisome block.

## Data Availability

Not applicable.

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
