# Peer review of "Consequences and Resolution of Transcription–Replication Conflicts"

_life, 2021, doi:10.3390/life11070637_

Round 1

Reviewer 1 Report

This review overall has a good quality and to certain extend, indeed offers the reader quite a good amount of information with regards to the topic of transcription-replication conflicts. The general feeling for the writing style is a little bit of mixed and messy, and the points that authors want to make were not well shown in a nice logical order. At least, I suggest using subtitles and subpoints in each big section to highlight the central ideas. Also, the figures are somewhat disconnected with the flow of the main text.

  1. Does the author believe that the Part 2 (methods to investigate TRCs) belong to this current spot or even belong to this manuscript?

  1. The format of paragraphs needs to be adjusted. Between some paragraphs you have space and some there is no extra space. In some paragraphs, texts were aligned to the left side, and some were aligned to both sides. Also, fonts were used quite messy through the whole manuscript.

  1. Cartoon figures need to be clearly labeled on the side what they are representing. For example, in Figure 1, what’s the blue module versus the orange module? This suggestion applies to all the figures and especially, in Figure 2, what’s the pink module versus the orange module?

Author Response

We thank the reviewer for her/his overall positive assessment of our manuscript. We have now carefully revised all chapters to provide a less mixed writing style and a more logical overview on the topic of transcription-replication conflicts. As also pointed out by reviewer 3, we realize that our choice to separate the methods to investigate TRCs as an independent chapter caused many avoidable redundancies, and we now followed the reviewer’s suggestion to merge chapters 2 and 3, focusing on the genetic consequences of TRCs in different model systems/organisms.

Together with this reorganization and the addition of subtitles and subpoints in each big section, we also made major changes to the figures (e.g. combining old Figures 2 and 3 to a new Figure 2), but also revised all other figures in order to connect them better with the flow of the main text.

Does the author believe that the Part 2 (methods to investigate TRCs) belong to this current spot or even belong to this manuscript?

The choice to separate the methods to investigate TRCs as an independent chapter was initially motivated by the fact that this specific topic has - to our knowledge – not been specifically reviewed and we considered this as a gap in the otherwise rich review literature on this topic. However, we agree that this caused many avoidable redundancies, and we followed the reviewer’s suggestion to merge chapter 2 and chapter 3, focusing on the genetic consequences of TRCs in different model systems/organisms.

The format of paragraphs needs to be adjusted. Between some paragraphs you have space and some there is no extra space. In some paragraphs, texts were aligned to the left side, and some were aligned to both sides. Also, fonts were used quite messy through the whole manuscript.

We apologize for these erroneous formatting errors that are now corrected in the revised version of the manuscript.

Cartoon figures need to be clearly labeled on the side what they are representing. For example, in Figure 1, what’s the blue module versus the orange module? This suggestion applies to all the figures and especially, in Figure 2, what’s the pink module versus the orange module?

We thank the reviewer for her/his helpful suggestions. We have revised all figures accordingly and now provide clearer labels in the figures as well as more detailed explanations on the different modules in the figure legends.

Reviewer 2 Report

Very nice and comprehensive review written in a very logically organized and easy to follow way. I am just not very happy with the figures. In figure 1 abbreviations are used that have not yet been introduced in the text before referencing to figure 1. But apart from that, it would facilitate reading in general, if the abbreviations would be explained at the end of each figure legend in any case. And also a brief color code explanation would make it more comfortable.

Author Response

Very nice and comprehensive review written in a very logically organized and easy to follow way. I am just not very happy with the figures. In figure 1 abbreviations are used that have not yet been introduced in the text before referencing to figure 1. But apart from that, it would facilitate reading in general, if the abbreviations would be explained at the end of each figure legend in any case. And also a brief color code explanation would make it more comfortable.

We thank the reviewer for her/his overall positive assessment of our manuscript. In alignment with all other reviewer’s comments, we have now carefully revised all figures of the manuscript and now provide clearer labels in the figures as well as more detailed explanations on the different modules and abbreviations in the figure legends.

Reviewer 3 Report

In this manuscript, Lalonde and colleagues from the Hamperl lab review the literature on transcription-replication conflicts considering the genetic and epigenetic levels.

Globally, I found that there is a lack of homogeneity and balance between the different parts. It appears that the different parts have been written separately by the first authors and the choice to separate the methods from the genetic consequences and the resolution mechanism introduces a lot of redundancies

The section 1 is not a section per se but rather an introduction to TRCs. The title should be removed.

  • Transcriptionnal R-loops must be defined in this section as a possible cause of TRCs. R-loops are mentioned in the following sections without being previously described in details.
  • I do not see why the replication of G-MIDS in G2 would represent a failsafe mechanism to avoid TRCs because there is still transcription during the G2 phase. Indeed, BIR is sensitive to HO TRCs (Liu et al., Nature 2021).

Even MIDAS occurring during mitosis, when transcription is globally shut down, may occur on partially condensed chromatin that would also be permissive for transcription.

  • From what I understood in ref [167], the authors proposed that Fork cleavage/relegation cycles would occur during S phase. The figure 1 describes that it occurs during mitosis.

The section 2 describes the methods to investigate TRCs.

I understand that the authors wanted to separate the methods from the genetic consequences but I feel that it would have been better to describe the methods as well as the main results gathered in different organisms (common results and differences), including the consequences on genomic instability. I would thus suggest to fuse sections 2 and 3, part by part:

in vivo artificial TRCs + TRC-induced mutations and other DNA alterations (recombination)

in vivo endogenous TRCs + CFSs and ERFs instability and other DNA alterations

  • For the in vitro part, Pomerantz and O’Donnell investigated CD and HO conflicts (refs [26] and [29]). It would be better to cite them together. The description of the mechanisms of restart proposed in these articles could be kept for the resolution section.
  • Also, regarding the suggestion line 137, Schauer et al., PNAS 2020 investigated the consequence of protein-bound RNA/DNA hybrids on replication progression in vitro.

  • in vivo artificial part, lines 167-168: “the orientation seems to determine how these events are sensed”. More precisely, the orientation seems to determine how the events are processed, which will influence the corresponding DNA damage response. Moreover, it would be better to underline that this difference in sensing has only been detected (evaluated?) with the episomal system in human cells.

  • in vivo endogenous part, lines 192-193: the work by the Posas lab (ref [168]) should be cited here and removed from the conclusion part.

  • Replication stress-induced TRCs should be described here in greater details. This could include ERFs and CFSs. Description of Ref [146] could be included because it describes replication pausing at a natural HO TRC in WT cells treated with HU. Oncogene-induced TRCs (ref [21]) should also be described in details in this section
  • Line 195 about Top1 inhibition, I would suggest to keep this approach for the section about chromatin changes (4.3).
  • About the PLA PCNA-RNAP II, a major drawback of this approach is that PCNA is also needed for many DNA repair reactions that require DNA synthesis and thus does not undoubtedly indicate a TRC
  • Lines 208-209, sites of EdU incorporation not only indicate active replication forks but more globally sites of nascent DNA synthesis. SIRF could then indicate the proximity of a protein with nascent DNA behind the fork.

The section 3 is dedicated to described the genetic consequences of TRCs.

As proposed above, parts of this section should be fused to the parts of section 2.

  • In 3.1, too few information is given about the genetic changes induced by artificial TRCs in yeast (on plasmids and chromosomes). For example, ref [35] is cited in section 2 without describing the main results.

  • On the contrary in 3.2, too much emphasis is made on CFSs when very few evidence points to the occurrence of TRCs at CFSs. Indeed, after dedicating a one-page section on the definition of CFSs and their fragility upon replication stress, the authors refer to the work of Brison et al., Nat Com 2019 [65] that concludes that CFS fragility is not related to transcription. Moreover, the authors forget to mention the work of Blin et al., NSMB 2019, which shows that elevating transcription at CFSs even decrease their fragility.

I would advise the authors to mention in a more synthetic manner the debate about the occurrence of TRCs at CFSs and that further investigations are needed to clarify it.

  • The first paragraph of 3.3 is still about CFSs. This information should be homogenized with 3.2
  • Line 358-359, the authors should not mix all systems to induce replication stress through the overexpression of oncogenes in the same bag. Ref [21] proposed that TRCs occur in their specific system and ref [80] described that DNA alteration occur in CFSs in another system. These references should not be used to propose that TRCs are occurring in CFSs.
  • Lines 377-378, regarding ERFs, the authors should be careful about the interpretation of ref [82]. Transcription of ERFs has not been specifically assessed in early S phase. Hence, TRCs cannot be considered as potent drivers of the instability at ERFs.
  • Refs [13] and [143] are complementary to [82], showing that TRCs (R-loops and P-RPA) preferentially occurs at HO genes that are replicated in early S phase.

Section 4

I feel that dedicating a section to the consequences of TRCs on the epigenome is not justified because this aspect is still very speculative. It would be better to mention the consequences of TRCs on the epigenome as an open question. Indeed, it is redundant to the conclusion part.

In fact, this section is rather dedicated to describe the changes in chromatin that may impact on TRCs.

4.1 histone marks that may prevent TRCs. In the same line as ref[94], the work of Frenkel et al., Genome Res 2021 proposes that H3 acetylation is slowing down replication to prevent genomic instability.

4.2 changes in chromatin remodeling that foster R-loop formation and may induce genomic instability.

4.3 topology and compaction of chromatin

            - works from refs at line 643 should be described in greater details in this section instead of being quickly cited in 5.3. It should also include ref [15]

Section 5 is dedicated to the resolution of TRCs

The figure 4 in this part is particularly unclear.

  • In the legend, (A), (B), (C), etc must come before the description
  • In figure 4B, RNAP II seems to be degraded while it is indicated to be skipped
  • What do the black and red arrows mean in every panel???
  • What is the scheme above (D) and (E) about???
  • (D) there is no way that one can understand how the replisome can skip the RNAP II with this scheme
  • (E) the scheme is erroneous. After the fork cleavage, the RNAP II cannot restart by using the broken DNA fragment as a template. See [167].

  • In 5.1, the authors propose that RNAP skipping and DNA synthesis repriming could be a mechanism for TRC resolution. How would the CMG replicative helicase skip the RNAP II complexes? This is conceivable for skipping a DNA lesion or even a small DNA-protein crosslink (Sparks et al., Cell 2019) but not for big RNAP II complexes. This should be discussed.
  • Lines 544-545, in a HO TRC, the mRNA is used as a primer for leading strand synthesis [26].

  • In 5.2, the authors propose that the Rad26-Rsp5-Def1 pathway could remove the RNAP II in a TRC but this has been tested in [129] and does not seem to be the case. I propose to skip this hypothesis.

  • In 5.3, the authors forgot to mention XPG and XPF nucleases as a R-loop processing pathway (Sollier et al., Mol Cell 2014).
  • The paragraph from lines 678 to 691 is imprecise and misleading. What are SMARCAL1 (not SMARCAD1) and ZRANB3? Fork reversal would allow the replication to go backward and may help RNAP II removal before replication restart. I do not see how template-switching, which is mainly a mechanism of post-replicative repair of lesions within ssDNA gaps, could be coupled to fork reversal to skip RNAP II. The scheme in figure 4D does not help to understand. Remove?
  • The cleavage and relegation cycles mechanism is the only one proposing that transcription should have the priority for the template and should be outlined. However, it is difficult to envision that the forks would undergo such cycles at every TRC to promote replication progression, knowing that a DSB is one of the most deleterious DNA lesions.

In conclusion, I strongly encourage the authors to re-organize the different sections to gain in clarity and avoid redundancies. I hope that the suggestions that I have made will help them in this process.

Author Response

In this manuscript, Lalonde and colleagues from the Hamperl lab review the literature on transcription-replication conflicts considering the genetic and epigenetic levels.Globally, I found that there is a lack of homogeneity and balance between the different parts. It appears that the different parts have been written separately by the first authors and the choice to separate the methods from the genetic consequences and the resolution mechanism introduces a lot of redundancies

We thank the reviewer for her/his constructive feedback and very much appreciate the many great suggestions and comments to help us improving our manuscript. We have now carefully revised and homogenized all chapters to provide a more balanced and less redundant overview on the topic of transcription-replication conflicts, as also pointed out by the other reviewer. The choice to separate the methods to investigate TRCs as a separate chapter was initially motivated by the fact that this specific topic has - to our knowledge – not been specifically addressed and we considered this as a gap in the otherwise rich review literature on this topic. However, we agree that this caused many avoidable redundancies, and we followed the reviewer’s suggestion to merge chapter 2 and chapter 3 (see more specific comments below). 

The section 1 is not a section per se but rather an introduction to TRCs. The title should be removed.

We agree and removed the title of the Introduction paragraph.

Transcriptionnal R-loops must be defined in this section as a possible cause of TRCs. R-loops are mentioned in the following sections without being previously described in details.

We thank the reviewer for this suggestion. We now introduce R-loops as a possible cause of TRCs at the beginning of revised chapter 2 where we discuss in detail the different types of co-transcriptional obstacles that can block replication fork progression and lead to TRCs. We hope the reviewer agrees with this choice. 

I do not see why the replication of G-MIDS in G2 would represent a failsafe mechanism to avoid TRCs because there is still transcription during the G2 phase. Indeed, BIR is sensitive to HO TRCs (Liu et al., Nature 2021). Even MIDAS occurring during mitosis, when transcription is globally shut down, may occur on partially condensed chromatin that would also be permissive for transcription.

We apologize for not being clearer here. We did not mean to suggest that G-MiDS is a failsafe mechanism to avoid TRCs, but rather a failsafe mechanism to complete DNA synthesis and therefore avoid deleterious consequences from such G-MiDS hotspots. In the cited paper (Wang et al., Cell Reports 2021), the authors show that certain TSS remain persistently under-replicated in S-phase due to RNAP complexes at the TSS that block replication fork progression. Despite ongoing transcription in G2, the authors show that these particular gaps of under-replicated DNA are indeed filled only in G2/M phase cells, suggesting that G-MiDS is a gap filling mechanism that reassures complete genome duplication prior to the onset of mitosis. We have tried to clarify this better in the text.

From what I understood in ref [167], the authors proposed that Fork cleavage/relegation cycles would occur during S phase. The figure 1 describes that it occurs during mitosis.

The reviewer is absolutely right and we apologize for this oversight, we have removed this from figure 1.

The section 2 describes the methods to investigate TRCs

I understand that the authors wanted to separate the methods from the genetic consequences but I feel that it would have been better to describe the methods as well as the main results gathered in different organisms (common results and differences), including the consequences on genomic instability. I would thus suggest to fuse sections 2 and 3, part by part

in vivo artificial TRCs + TRC-induced mutations and other DNA alterations (recombination)

in vivo endogenous TRCs + CFSs and ERFs instability and other DNA alterations

We thank the reviewer for this great suggestion and totally agree that our initial choice to separate the methods from genomic consequences created many avoidable redundancies in the text. In the revised version, we followed the reviewer’s suggestion and combined sections 2 and 3 with a focus on the effects of TRCs on genetic stability at (2.1) artificial reporter and plasmid constructs and (2.2) at endogenous chromosomal loci. 

For the in vitro part, Pomerantz and O’Donnell investigated CD and HO conflicts (refs [26] and [29]). It would be better to cite them together. The description of the mechanisms of restart proposed in these articles could be kept for the resolution section.

Thank you for this suggestion. We have removed the in vitro part and describe these results now together in the resolution section.

Also, regarding the suggestion line 137, Schauer et al., PNAS 2020 investigated the consequence of protein-bound RNA/DNA hybrids on replication progression in vitro.

We added the work of Schauer et al. PNAS 2020 at this section. Thank you for this excellent suggestion.

 in vivo artificial part, lines 167-168: “the orientation seems to determine how these events are sensed”. More precisely, the orientation seems to determine how the events are processed, which will influence the corresponding DNA damage response. Moreover, it would be better to underline that this difference in sensing has only been detected (evaluated?) with the episomal system in human cells.

We have rephrased this sentence accordingly. We also provide a much more critical discussion on these results: “- it is important to note that events on these short (~ 10 kb) plasmids may differ in for example topological aspects and not accurately model all of the chromatin dynamics and endogenous chromosomal transactions in the genome. Thus, it will be important to evaluate results from the episomal system regarding for example the orientation-dependent DNA damage responses in a more physiological context.”

in vivo endogenous part, lines 192-193: the work by the Posas lab (ref [168]) should be cited here and removed from the conclusion part.

We followed the reviewer’s suggestion, thanks!

Replication stress-induced TRCs should be described here in greater details. This could include ERFs and CFSs. Description of Ref [146] could be included because it describes replication pausing at a natural HO TRC in WT cells treated with HU. Oncogene-induced TRCs (ref [21]) should also be described in details in this section

We have now included and discuss these references in section 2.2.3

Line 195 about Top1 inhibition, I would suggest to keep this approach for the section about chromatin changes (4.3).

We followed the reviewer’s suggestion, thanks!

About the PLA PCNA-RNAP II, a major drawback of this approach is that PCNA is also needed for many DNA repair reactions that require DNA synthesis and thus does not undoubtedly indicate a TRC

We agree with the reviewer. As the focus of the new combined section is no longer on the methods to investigate TRCs, we have removed the description of the PLA approach from the text and the figure.

Lines 208-209, sites of EdU incorporation not only indicate active replication forks but more globally sites of nascent DNA synthesis. SIRF could then indicate the proximity of a protein with nascent DNA behind the fork

We agree with the reviewer. As the focus of the new combined section is no longer on the methods to investigate TRCs, we have removed the description of the SIRF approach from the text.

The section 3 is dedicated to described the genetic consequences of TRCs.

As proposed above, parts of this section should be fused to the parts of section 2.

We combined the two sections, as mentioned above. Thank you.

In 3.1, too few information is given about the genetic changes induced by artificial TRCs in yeast (on plasmids and chromosomes). For example, ref [35] is cited in section 2 without describing the main results.

We agree with the reviewer and now expand this section to provide a more detailed and balanced description on the results from artificial TRCs in yeast (section 2.1.2)

 On the contrary in 3.2, too much emphasis is made on CFSs when very few evidence points to the occurrence of TRCs at CFSs. Indeed, after dedicating a one-page section on the definition of CFSs and their fragility upon replication stress, the authors refer to the work of Brison et al., Nat Com 2019 [65] that concludes that CFS fragility is not related to transcription. Moreover, the authors forget to mention the work of Blin et al., NSMB 2019, which shows that elevating transcription at CFSs even decrease their fragility. I would advise the authors to mention in a more synthetic manner the debate about the occurrence of TRCs at CFSs and that further investigations are needed to clarify it.

We agree with the reviewer and have now significantly shortened this section on CFS, at the same time expanding on the work in prokaryotes and yeast systems to give a more balanced view on the effects of TRCs on genomic instability in multiple model organisms. Importantly, we also include the work of Blin et al., NSMB 2019 and provide an overall much more synthetic debate on the occurrence of TRCs at CFSs. Certainly, further investigations are needed to clarify it. 

The first paragraph of 3.3 is still about CFSs. This information should be homogenized with 3.2

Done.

Line 358-359, the authors should not mix all systems to induce replication stress through the overexpression of oncogenes in the same bag. Ref [21] proposed that TRCs occur in their specific system and ref [80] described that DNA alteration occur in CFSs in another system. These references should not be used to propose that TRCs are occurring in CFSs.

We acknowledge the concerns of the reviewer regarding the conclusions drawn from Ref [21] and [80] and now indicate that different model systems of oncogene induction were used in both studies and that follow up investigations will be required to test whether oncogene-induced TRCs directly occur at CFS and induce their fragility.

Lines 377-378, regarding ERFs, the authors should be careful about the interpretation of ref [82]. Transcription of ERFs has not been specifically assessed in early S phase. Hence, TRCs cannot be considered as potent drivers of the instability at ERFs.

We acknowledge the reviewers concerns regarding the interpretation of Ref [82]. While transcription of ERFSs has indeed not been directly analyzed in early S-phase cells, the combined observation that  inhibition of transcription decreases ERFS fragility and at the same time oncogenic replication stress is a driver of this instability, an involvement of TRCs can be hypothesized. This view has also been suggested by Ref [83]. Agreeing with the reviewer that truly reliable data on this aspect is still lacking, we emphasized the incomplete data situation and rephrased our interpretation to properly indicate its partially speculative nature as well as a need for further investigation.

Refs [13] and [143] are complementary to [82], showing that TRCs (R-loops and P-RPA) preferentially occurs at HO genes that are replicated in early S phase.

We discuss these references now together in section 2.2.2

Section 4

I feel that dedicating a section to the consequences of TRCs on the epigenome is not justified because this aspect is still very speculative. It would be better to mention the consequences of TRCs on the epigenome as an open question. Indeed, it is redundant to the conclusion part.

In fact, this section is rather dedicated to describe the changes in chromatin that may impact on TRCs.

We thank the reviewer for raising this important point and agree that the usage of the term “epigenetic consequences” is misleading as it implies a heritable change by TRCs where currently little evidence exists. In the revised version, we only discuss the consequences of TRCs on the epigenome as an open question in the conclusion section. We followed the reviewer’s suggestion and changed the title of this section to describe the impact of the chromatin environment on TRCs.

4.1 histone marks that may prevent TRCs. In the same line as ref[94], the work of Frenkel et al., Genome Res 2021 proposes that H3 acetylation is slowing down replication to prevent genomic instability.

We have now included and discuss this reference. Thank you!

4.2 changes in chromatin remodeling that foster R-loop formation and may induce genomic instability.

We made minor changes to this section and now highlight better the potential connection of chromatin remodeling and R-loop formation.

4.3 topology and compaction of chromatin

            - works from refs at line 643 should be described in greater details in this section instead of being quickly cited in 5.3. It should also include ref [15]

We have included and discuss the results from reference 15 in this section. Regarding the references from line 643, our purpose to combine these citations was to point out the abundancy (and potentially) redundancy of this growing list of DEAD-box RNA helicases that can unwind RNA:DNA hybrid structures in cells. Since the direct link between this family of helicases and torsional stress relaxation of DNA remains to be shown, the authors feel that it is best to not include them in this sub-section.

Section 5 is dedicated to the resolution of TRCs

The figure 4 in this part is particularly unclear.

We agree and apologize for the lack of clarity in this figure. The conversion of the picture from .svg to .png altered the figures and arrows were left with arrowheads only without the line, making the figure unreadable. We are sorry for this overlook and corrected it while updating it with the reviewer’s comments below.

Following the comment of the reviewer on template switching, we altered the scheme to show the fork reversal as a stabilization to give time to resolve the conflict instead of talking about template switching.

In the legend, (A), (B), (C), etc must come before the description

We have changed the order.

In figure 4B, RNAP II seems to be degraded while it is indicated to be skipped

We agree that this part of the figure was unclear. The intention was to show both CD-RNAP skipping, and RNA takeover, which imply the eviction of the RNAP. To clarify, we merged HO- and CD-RNAP skipping and gave the RNA takeover process its own panel.

What do the black and red arrows mean in every panel???

The black arrows shows either the different mechanisms or steps and the red arrows depict movement of the complexes. It is more evident with the actual arrows and not just the arrowheads. This issue has been resolved and we hope the meaning of the arrows are clearer in the revised version of the figure.

What is the scheme above (D) and (E) about???

It should have represented the starting point, the TRC, from which the different TRC resolving mechanisms emerge. We hope that this will be clearer now with the arrows.

 (D) there is no way that one can understand how the replisome can skip the RNAP II with this scheme

Following the comment of the reviewer on template switching, we altered the scheme to show the fork reversal as a mechanism to stabilize the replisome and therefore give more time to resolve the conflict instead of talking about template switching in this context.

 (E) the scheme is erroneous. After the fork cleavage, the RNAP II cannot restart by using the broken DNA fragment as a template. See [167].

We have improved the scheme and hope it's more clear now. We have also tried to explain this better in the text.

In 5.1, the authors propose that RNAP skipping and DNA synthesis repriming could be a mechanism for TRC resolution. How would the CMG replicative helicase skip the RNAP II complexes? This is conceivable for skipping a DNA lesion or even a small DNA-protein crosslink (Sparks et al., Cell 2019) but not for big RNAP II complexes. This should be discussed.

We agree with the reviewer’s concern. It has been shown that the CMG helicase can bypass interstrand crosslinks and DNA-protein crosslinks but it is true that larger RNAP complexes have not been assessed in those studies. We added a paragraph to describe the current state of knowledge on this matter and formulate this as an open question.

Lines 544-545, in a HO TRC, the mRNA is used as a primer for leading strand synthesis [26].

Pomerantz and O’Donnell shows that the RNA is used as a primer for leading strand synthesis in a CD-TRC, but not in a HO-TRC. In a HO-TRC, we cannot envision how the 5’-3’ orientation of the RNA will allow it to be used as a primer for DNA synthesis.

In 5.2, the authors propose that the Rad26-Rsp5-Def1 pathway could remove the RNAP II in a TRC but this has been tested in [129] and does not seem to be the case. I propose to skip this hypothesis.

In this article, Poli and co-workers have demonstrated that the Rad26-Rsp5-Def1 pathway is not required to degrade RNAPs stalled under conditions of HU-induced replication stress. Nonetheless, we think that TC-NER represents an important repair pathway allowing the eviction or restart of RNAP complexes that would otherwise block a travelling replisome. We modified the text to emphasize that it is not a mechanism specifically occurring at TRC sites but a mechanism that can clear the path for the replisome before the collision occur. We also modified the text to emphasize the non-overlapping functions of those two mechanisms.

In 5.3, the authors forgot to mention XPG and XPF nucleases as a R-loop processing pathway (Sollier et al., Mol Cell 2014).

Thanks for noticing this oversight. We now include and discuss this reference.

The paragraph from lines 678 to 691 is imprecise and misleading. What are SMARCAL1 (not SMARCAD1) and ZRANB3? Fork reversal would allow the replication to go backward and may help RNAP II removal before replication restart. I do not see how template-switching, which is mainly a mechanism of post-replicative repair of lesions within ssDNA gaps, could be coupled to fork reversal to skip RNAP II. The scheme in figure 4D does not help to understand. Remove?

As suggested, we removed the section on template switching and simplified the description of fork reversal. Furthermore, we refer the reader to a recent review on fork reversal as the added details on fork reversal processes complicate this section and are not needed for the main concept that fork reversal is a mechanism to stabilize the replisome and therefore give more time to resolve the conflict.

The cleavage and relegation cycles mechanism is the only one proposing that transcription should have the priority for the template and should be outlined. However, it is difficult to envision that the forks would undergo such cycles at every TRC to promote replication progression, knowing that a DSB is one of the most deleterious DNA lesions.

We modified the text accordingly. It is indeed an important fact to outline.

In conclusion, I strongly encourage the authors to re-organize the different sections to gain in clarity and avoid redundancies. I hope that the suggestions that I have made will help them in this process.

We thank the reviewer for her/his constructive suggestions that have helped us tremendously to improve our manuscript. Thank you very much.

Round 2

Reviewer 3 Report

 I acknowledge that the authors have spent many efforts to implement all the suggestions I have made to improve their manuscript and congratulate them for the great work.

I have still few minor corrections to propose.

  • Line 109: The definition of R-loops is incorrect. R-loops are not “non-productive configurations of RNAPII”. These structures form naturally during transcription and can play many physiological roles, for example during transcription termination (see Garcia-Muse & Aguilera, Cell 2019). They could also represent a threat to genomic instability, possibly in TRCs.

  • Section 2.2.1. The controversy about the role of highly transcribed genes impeding DNA replication in eukaryotes could be cited (Azvolinsky et al., Mol Cell 2009; Tran et al., Nat Commun 2017 versus Osmundson et al., Nat Struct Mol Biol 2017; Yeung & Smith, Genetics 2020).

  • Section 2.2.3 should not be limited to "mammalian genomes" but to eukaryotic genomes. Indeed, the work by the Posas lab, which should be discussed more extensively and include the first publication (Duch et al., Nature 2013), has been done in budding yeast. Also in yeast, the work by Hoffman et al., Genome Research 2015 shows that hydroxyurea treatment induces transcriptional changes that may cause TRCs.

  • Section 4.3: Among RNA:DNA helicases that could remove R-loops, Senataxin is the best studied and could therefore be described in greater details. For example, in its absence, replication is strongly blocked by HO TRCs (Alzu et al., Cell 2012) and the activation of dormant origin is required to complete replication (Brambati et al., NAR 2018). R-loops are presumably processed into DSBs in Senataxin-depleted cells (Sollier et al., Mol Cell 2014). It has also recently been shown that Sen1 is specifically recruited to replication forks (Appanah et al., Cell Reports 2020).

  • Figure 4 (E): the scheme is still wrong. The RNAPII cannot restart by using the broken sister-chromatid as a template (RNAP passage). See the original model published by Chappidi et al., Mol Cell 2020.

Author Response

I acknowledge that the authors have spent many efforts to implement all the suggestions I have made to improve their manuscript and congratulate them for the great work.

We thank the reviewer for her/his positive feedback and we are glad that we could address the comments and improve the manuscript with the help of the reviewer.

I have still few minor corrections to propose.

Line 109: The definition of R-loops is incorrect. R-loops are not “non-productive configurations of RNAPII”. These structures form naturally during transcription and can play many physiological roles, for example during transcription termination (see Garcia-Muse & Aguilera, Cell 2019). They could also represent a threat to genomic instability, possibly in TRCs.

Thank you for pointing this out. We rephrased the text accordingly (lines 107-115): For example, RNAP complexes can be paused proximal to promoters, productively elongating along the gene body, or assume other configurations such as backtracking or the formation of R-loops. R-loops are three-stranded secondary DNA structures, where the nascent RNA strand rehybridizes with the complementary DNA template strand, resulting in an RNA:DNA hybrid plus displaced single-stranded DNA [24]. Although these structures form naturally during transcription and have been ascribed many physiological roles in cellular processes (reviewed in [25]), the presence of R-loops in the context of TRCs is thought to stall transcription ahead of the replisome and thereby have a negative impact on genome stability.

Section 2.2.1. The controversy about the role of highly transcribed genes impeding DNA replication in eukaryotes could be cited (Azvolinsky et al., Mol Cell 2009; Tran et al., Nat Commun 2017 versus Osmundson et al., Nat Struct Mol Biol 2017; Yeung & Smith, Genetics 2020).

Thank you for this great suggestion. We changed the title of section 2.2.1 and removed “on bacterial chromosomes” and now include the discussion on the role of highly transcribed genes in eukaryotes as suggested.

Section 2.2.3 should not be limited to "mammalian genomes" but to eukaryotic genomes. Indeed, the work by the Posas lab, which should be discussed more extensively and include the first publication (Duch et al., Nature 2013), has been done in budding yeast. Also in yeast, the work by Hoffman et al., Genome Research 2015 shows that hydroxyurea treatment induces transcriptional changes that may cause TRCs.

We have changed the title of this section to eukaryotic genomes and now describe in more detail the results obtained from (Duch et al., Nature 2013 and Hoffman et al., Genome Research, 2015). Thank you for this great suggestion.

Section 4.3: Among RNA:DNA helicases that could remove R-loops, Senataxin is the best studied and could therefore be described in greater details. For example, in its absence, replication is strongly blocked by HO TRCs (Alzu et al., Cell 2012) and the activation of dormant origin is required to complete replication (Brambati et al., NAR 2018). R-loops are presumably processed into DSBs in Senataxin-depleted cells (Sollier et al., Mol Cell 2014). It has also recently been shown that Sen1 is specifically recruited to replication forks (Appanah et al., Cell Reports 2020).

Thank you very much for this suggestion, we have now expanded this section and describe the role of Senataxin in removing R-loops in greater detail.

Figure 4 (E): the scheme is still wrong. The RNAPII cannot restart by using the broken sister-chromatid as a template (RNAP passage). See the original model published by Chappidi et al., Mol Cell 2020.

We apologize for this oversight, we have now corrected the figure according to Chappidi et al., Mol Cell 2020.

Thank you very much!